# Disruptions of Circadian Genes in Cutaneous Melanoma—An In Silico Analysis of Transcriptome Databases

**DOI:** 10.3390/ijms241210140

**Published:** 2023-06-14

**Authors:** Monika Lesicka, Bogusław Nedoszytko, Edyta Reszka

**Affiliations:** 1Department of Translational Research, Nofer Institute of Occupational Medicine, 91-349 Lodz, Poland; monika.lesicka@imp.lodz.pl; 2Department of Dermatology, Venerology and Allergology Medical University of Gdansk, 80-211 Gdansk, Poland; boguslaw.nedoszytko@gumed.edu.pl; 3Molecular Laboratory, Invicta Fertility and Reproductive Centre, Polna 64, 81-740 Sopot, Poland

**Keywords:** skin cutaneous melanoma, circadian genes, immune infiltration level, bioinformatic analysis

## Abstract

Circadian genes are a set of genes that regulate the body’s internal clock and influence various physiological processes, including sleep–wake cycles, metabolism and immune function. Skin cutaneous melanoma (SKCM) is a type of skin cancer that arises from the pigment-producing cells in the skin and is the most deadly form of skin cancer. This study has investigated the relevance of circadian gene expression and immune infiltrations in the outcomes of cutaneous melanoma patients. In the present study, in silico methods based on the GEPIa, TIMER 2.0 and cBioPortal databases were performed, so as to investigate the transcript level and prognostic value of 24 circadian genes in SKCM and their relationship with the immune infiltration level. The in silico analysis showed that significantly more than half of the investigated circadian genes have an altered transcript pattern in cutaneous melanoma compared to normal skin. The mRNA levels of *TIMELES* and *BHLHE41* were upregulated, whereas those of *NFIL3*, *BMAL1*, *HLF*, *TEF*, *RORA*, *RORC*, *NR1D1*, *PER1*, *PER2*, *PER3*, *CRY2* and *BHLHE40* were downregulated. The presented research shows that SKCM patients with at least one alteration of their circadian genes have decreased overall survival. Additionally, majority of the circadian genes are significantly corelated with the immune cells’ infiltration level. The strongest correlation was found for neutrophils and was followed by circadian genes: *NR1D2* r = 0.52 *p* < 0.0001, *BMAL1* r = 0.509 *p* < 0.0001; *CLOCK* r = 0.45 *p* < 0.0001; *CSNKA1A1* r = 0.45 *p* < 0.0001; *RORA* r = 0.44 *p* < 0.0001. The infiltration level of immune cells in skin tumors has been associated with patient prognosis and treatment response. Circadian regulation of immune cell infiltration may further contribute to these prognostic and predictive markers. Examining the correlation between circadian rhythm and immune cell infiltration can provide valuable insights into disease progression and guide personalized treatment decisions.

## 1. Introduction

Cutaneous melanoma (SKCM) is a type of skin cancer arising from the melanocytes, which has demonstrated increasing incidence rates over the past few decades. It is considered to be one of the most aggressive forms of skin cancer and is responsible for the majority of skin cancer-related deaths [1]. The highest incidence rates of cutaneous melanoma have been reported in Australia, followed by the southern states of the United States [2]. Study of the circadian rhythm in cutaneous melanoma is an area of research that has gained considerable attention in recent years [3,4]. This is because disruptions in the circadian rhythm have been linked to an increased risk of cancer and it is believed that understanding the circadian rhythm in cutaneous melanoma may lead to the new prevention and treatment strategies [5].

Every organism needs to maintain homeostasis to function properly. Circadian rhythm is the basic process that helps maintain it. Over the course of evolution and adaptation to the external environment, all organisms on Earth have developed a mechanism that regulates the rhythms of all physiological processes [6]. Circadian rhythm is a fundamental process that organizes the functioning of an organism in response to external stimuli. This adaptation process is observed in the entire organism at a physiological level, as well as peripherally—at the levels of tissues and cells—by influencing various molecular pathways regulated by circadian genes transcriptional–translational feedback loops, such as metabolism, cell proliferation, inflammation and DNA damage repair [6].

Molecular clocks are controlled by multiple genes that form basal and auxiliary transcriptional and translational feedback loops with both positive and negative characteristics. To date, approximately 20 candidate genes related to the circadian rhythm generation and maintenance in the hippocampus and suprachiasmatic nucleus (SCN), as well as in peripheral tissues including skin, have been identified (Appendix A). The circadian-driven feedback loop consists of transcription-inducing complexes of the negative clock regulators PER and CRY, along with other positive factors BMAL1/CLOCK or BMAL1/NPAS2. In the cytoplasm, PER and CRY are phosphorylated by CSNK1A1. Accumulated clock proteins translocate to the nucleus and inhibit its BMAL1 expression in the form of protein complexes. Other bHLH-PAS transcription factor family members—RORs (RAR-related orphan receptor), NR1Ds or REV-ERBs (nuclear receptor subfamily 1, group D members)—constitute the stabilizing loop by repressing BMAL1 transcription via retinoic acid-related orphan receptor response element (RORE) in the promoter region of BMAL1 [7,8]. The regulatory role (particularly of the BMAL1/CLOCK heterodimer) in controlling expression of various genes involved in multiple biological processes in a cell, such as genetic alterations in metabolism, inflammatory pathways, cell cycle arrest, DNA damage repair, cell proliferation, maintenance of genomic stability, oxidative stress and apoptosis, is a unique feature of circadian genes. Moreover, circadian genes also affect the expression of genes involved in skin barrier function, inflammation and response to UV radiation [9]. Perturbations in these processes are hallmarks of cancer including skin cancer, while chronic circadian rhythm disruption predisposes tumor development [7,8].

Inflammation is another factor that has been implicated in the development and progression of cutaneous melanoma. Chronic inflammation has been linked to an increased risk of cancer. It is also believed that inflammation plays a role in the growth and spread of melanoma cells. In fact, studies have found that melanoma cells can produce inflammatory cytokines that promote tumor growth and suppress the immune system.

The present article summarizes circadian gene alterations at the transcriptomic level and the impact of circadian rhythm on the immune infiltration level of cutaneous melanoma using publicly available transcriptomic databases.

## 2. Results

### 2.1. Gene Expression Level of the Core Circadian Genes in Skin Cutaneous Melanoma

Expression level of the 24 core circadian genes was obtained using the GEPIA database. The list of selected circadian genes is presented in Appendix A. We performed the in silico analysis of 24 circadian genes (Appendix A) in 558 cancerous tissues and 461 normal skin samples. We found that 12 of the 24 investigated genes were statistically down-expressed, including *NFIL3*; transcription factors *BMAL1*, *HLF*, *TEF*; nuclear receptor factors *RORA*, *RORC*, *NR1D1*; period genes *PER1*, *PER2*, *PER3*; cryptochrome *CRY2*; Basic Helix-Loop-Helix Family *BHLHE40*, *BHLHE41*, and two were overexpressed: *TIMELESS*, *BHLHE41* (Figure 1). Further analysis showed that 12 circadian genes, *BMAL1*, *CLOCK*, *CRY1*, *CRY2*, *HLF*, *NPAS2*, *NR1D2*, *PER1*, *PER2*, *PER3*, *TEF*, *TIMELESS*, were associated with tumor stage (Figure 2).

### 2.2. Survival Analysis According to Gene Expression Level of the Core Circadian Genes in Skin Cutaneous Melanoma

The search for overall survival or disease-free survival biomarkers in melanoma remains desirable. Therefore, we performed an analysis of the core circadian genes in the GEPIA database. We found that higher expression levels of BMAL1, NR1D2 and NR1D2 were associated with disease free survival (DFS). Opposite results were found for CSNKE1, where low expression at marginal significance was associated with DFS (Figure 3). Similar associations were found for the overall survival (OS), where significantly higher expression levels of NR1D2 and BMAL1 were associated with longer survival among skin cutaneous melanoma patients. Otherwise, lower expression levels of HLF and CSNK1E were significantly associated with OS among SKCM patients (Figure 4). The further Multivariate Cox regression analysis performed by the TIMER database showed that the origin of SKCM played a role. High hazard ratio of overall survival was linked to TIMELESS and RORA expression in primary cutaneous melanoma patients, with values of 3.05 (1.36–6.846) and 2.29 (1.195–4.37), respectively (Table 1).

### 2.3. Infiltration Level

Our study presents a correlation between circadian genes expression and immune infiltration in cutaneous melanoma. According to the in silico analysis performed using the TIMER database, there are many significant correlations between immune cell infiltration, including B cells, CD4+ T cells, CD8+ T cells, dendritic cells, macrophages and neutrophils, level and circadian gene expression levels in skin cutaneous melanoma patients. All the results are presented in Table 2 and Table 3. The strongest correlation between circadian gene expression and the level of immune cell infiltration was observed among neutrophils and *NR1D2*, r = 0.52, *p* < 0.0001; *BMAL1*, r = 0.509, *p* < 0.0001; *CLOCK*, r = 0.45, *p* < 0.0001; *CSNKA1A*, r = 0.45, *p* < 0.0001; *RORA*, r = 0.44, *p* < 0.0001. Additionally, we found an association between mutation of selected circadian genes and expression of *CTLA4*, *CD86*, *CD80* and *CD28*. We observed that patients with mutated *PER2* presented higher expression levels of crucial *CTLA4*, *CD86*, *CD80* and *CD28* (Figure 5). Moreover, *PER2* expression level was correlated with *CD86* and *CD28* expression levels (Figure 6B,C). Other interesting and significant correlations between *PDCD1* expression and *CRY2*, *BMAL1* and *CSNK1E* expression levels were r = 0.13, *p* = 0.04; r = 0.13, *p* = 0.04; r = −0.20, *p* = 0.0003, respectively (Figure 6A).

### 2.4. Mutations of Circadian Genes in Cutaneous Melanoma

According to the cBioPortal and TIMER 2.0 database, the most mutated circadian genes among cutaneous melanoma patients were PER1, PER2, RORB and RORC (Table 4). Additionally, these mutations were most frequent among skin cutaneous melanoma compared to other cancer patients. Types of mutations that occurred among circadian genes included inframe mutations, missense mutations, splice mutations, truncating mutation, structural mutation and amplification, as well as deep deletion (Figure 7). Some of these mutations had reflections in mRNA expression of circadian genes in cutaneous melanoma patients. Mutations of *BMAL1* and *NR1D2* were associated with the increased expression of their mRNA levels (Figure 8). The analysis of overall survival among cutaneous melanoma patients showed that the patients with at least one alteration within the selected circadian genes (24 genes) had a worse prognosis compared to the patients without any alterations in these genes (Figure 9). The unaltered group of patients lived 52 months longer than the altered group of patients (logrank test *p*-value = 0.03).

### 2.5. Methylation of Circadian Genes

Correlation between mRNA level of the selected circadian genes and methylation level was analyzed using thecBioPortal database. The strongest negative correlation between the transcription level and methylation was observed for PER3, RORC, CSNK1E and PER2. Otherwise, the strongest positive correlation was observed for RORA and HLF (Table 5). Moreover, we found that methylation of selected CpG sites of circadian genes was associated with overall survival of skin cutaneous melanoma patients. The in silico analysis showed that higher methylation levels of BHLHE41 (cg03046445), CLOCK (cg05960024), CSNK1E (cg01346718), CSNK1E (cg01441777), PER1 (cg16545079), PER2 (cg05664072), PER3 (cg25514503), RORC (cg18149207) and RORA (cg27167601) were correlated with longer overall survival (Figure 10). Moreover, significant relationships between methylation and OS were reported for CpG islands located in the transcription start site (TSS) (Table 5).

## 3. Discussion

The presented in silico research comprehensively analyzed the mRNA expression levels of 24 circadian genes and correlations with the immune infiltration level in skin cutaneous melanoma by the use of multipleomics databases.

Cutaneous melanoma is the leading cause of skin cancer deaths and is typically diagnosed at an advanced stage, which results in a poor prognosis. Therefore, new genetic targets for therapy and valuable prognostic biomarkers are desirable [10]. Our study showed disruptions of circadian genes in broad aspects. The presented in silico analysis showed that more than half of the investigated circadian genes represented significantly different expression patterns compared to normal skin. We found that *BHLHE41* and *TIMELESS* were upregulated in skin cutaneous melanoma patients, while *BMAL1*, *BHLHE40*, *CRY2*, *HLF*, *NFIL3*, *NR1D1*, *PER1*, *PER2*, *PER3*, *RORA*, *RORC* and *TEF* were downregulated. Moreover, a similar tendency for these genes was associated with more advanced stages (stages III and VI) of SCKM. Lengyel et al. have shown that mRNA levels of core circadian genes, including *PER1*, *PER2*, *CLOCK* and *CRY1*, were reduced in melanoma skin biopsies compared to adjacent normal skin, but in a limited number of patients [11]. Moreover, a later study conducted by de Assis et al. showed that patients with more advanced tumor stages (III and IV) had a lower *PER3* expression [12]. The presented in silico analysis of a larger group of patients partially proved that observation. We did not observe significant differences for mRNA levels of *CLOCK* and *CRY1* genes. An animal study conducted by de Assis et al. demonstrated that circadian gene expression was altered in melanoma compared to normal skin, pointing to the reduction in Per1 and Bmal1 in tumor melanoma-bearing mice [13]. Additionally, the same authors have shown that circadian genes were disrupted in metastatic cutaneous melanoma based on an in silico analysis [12]. We found a negative correlation between the transcript level of circadian genes and their methylation status, mostly in the case of CpG islands located in Transcription Start Site. An altered epigenetic pattern of circadian genes has been observed in cancer studies [14]. Methylation status of cfDNA cutaneous melanoma is considered as an early cancer detection biomarker. Our analysis showed that OS of SKCM patients was associated with methylation status of CpG site of circadian genes.

Additionally, mutations of circadian genes are also present in SKCM; the most mutated genes are *PER2* and *PER3,* which are classified as tumor suppressor genes [15,16,17]. Moreover, patients with at least one alteration among circadian genes have decreased overall survival. These observations may indicate that disruptions of circadian genes may play an essential role in the induction of cancer formation [18]. The link between circadian genes and cutaneous melanoma may be explained by the role of circadian genes in DNA damage repair, cellular proliferation and apoptosis [9]. In 2022, for example, a study by de Assis et al. found that increased *Bmal1* gene expression was associated with reduced melanoma cell proliferation [19]. Therefore, in silico analysis may be a powerful tool for investigating circadian genes dependent strategy for anticancer therapy of skin.

An observation suggesting that circadian genes may be used as potential prognostic biomarkers in SCFCM and are associated with poor prognosis in cutaneous melanoma patients (including disease-free survival (DFS) and overall survival (OS) rates) is another interesting result of the present study. A similar association has been found by de Assis et al., where *BMAL1* was positively correlated with overall survival in metastatic melanoma patients. High expression of *BMAL1* was associated with longer survival [12]. This observation was proven for a larger group of patients in our study. Other studies have already identified circadian genes as prognostic markers for other tumors. Triple-negative breast cancer patients with a high expression level of NR1D1 treated with chemotherapy had longer OS [20]. Our analysis showed that an increased *NR1D2* expression level was correlated with positive results, including OS and DFS, in SKCM patients. A study on stomach adenocarcinoma has revealed opposite results, where high NR1D1 and PER1 levels were correlated with poor OS [21]. Another study has also shown that decreased levels of PERs genes were associated with poor OS rates in non-small cell lung cancer patients [22]. Additionally, *CLOCK* has been linked with a worse prognosis for hepatocellular carcinoma [23].

Another interesting aspect of our study is related to the infiltration level of many immune cells, which is correlated with the majority of the investigated circadian genes. The tumor microenvironment (TME) plays a significant role in tumorigenesis and SKCM progression, but the dynamic regulation of the immune and stromal components is not yet fully understood [24]. The tumor microenvironment (TME) involves many types of cells, including tumor cells, fibroblasts, stromal cells, immune and inflammatory cells including natural killer cells (NK), tumor-associated macrophages (TAM), natural killer neutrophil T cells (NKT), dendritic cells (DC), T lymphocytes, B lymphocytes and the extracellular matrix (ECM). Immune cells in the TME are key factors affecting tumor growth and treatment response. The TME influences clinical outcomes and contains potential targets for therapeutic modulation [25]. In the present study, we quantified the correlation between circadian genes’ expression and the proportion of tumor-infiltrating immune cells, based on the TIMER 2.00 database [26,27].

Recently, increasing evidence has indicated that the tumor microenvironment (TME) is involved in tumor development. Interactions between cancer cells, stromal cells and tumor-infiltrating immune cells (TICs) are critical for malignant cancer progression, including promoting replicative immortality, invasion, metastasis and evasion of immune surveillance [28].

Multiple studies have reported that TIC represents a promising TME marker for assessing treatment effects. TIC components and their activation statuses are important parameters affecting patient prognosis and tumor characteristics. Anti-cytotoxic T lymphocyte antigen 4 (CTLA-4) therapy activates T cells and induces programmed death-ligand 1 (PD-L1) expression in tumor cells and TICs [29]. Activation of CD8+ T cells can prolong patients’ survival in many cancers, including SKCM [30]. A study has shown that increased CD8+ T cell trafficking contributes to the efficacy of anti-programmed death 1 (PD-1)/CTLA-4 therapy against melanoma metastasis and may represent an effective immunotherapeutic strategy [31]. Neutrophils also play a background role in melanoma and can actively switch to an antitumor mode [32]. Our analysis showed that the strongest correlation between circadian genes’ expression and immune infiltration was observed for neutrophils. The abovementioned studies demonstrate that crosstalk between cancer cells and the TME plays an integral role in the CM development, making it challenging to accurately characterize the dynamic regulation of the TME by immune and stromal components [29]. Furthermore, the circadian clock regulates differentiation of CD4(+) T helper [33]. We found a significant correlation between the CD4+ T cell infiltration level and mRNA expression of CRY2, DBP, PER2 and TEF (Table 4). The presented results also revealed that crucial cytokines, chemokine and other crucial immune biomarkers are significantly associated with circadian gene expression in both primary and metastatic melanoma (Table 3).

Aberrant inflammation, immunosuppression and evasion of immune surveillance promote tumorigenesis by increasing cell proliferation, cell survival, angiogenesis and invasiveness. Both the immune system and inflammatory processes are regulated by the circadian clock and its components [34,35]. NF-κB is one of the key regulators of inflammation, regulated by circadian clock components *BMAL1*, *CLOCK* and *CRY* [36]. *BMAL1* is physiologically downregulated during activation of NF-κB-related inflammatory responses through p65 subunit phosphorylation, but our analysis showed a positive correlation between the expression of NF-κB and *BMAL1* in both primary and metastasis cutaneous melanoma. On the other hand, *CLOCK* binds to p65 and increases NF-κB-induced transcription of inflammatory genes. CRY regulates NF-κB activation by modulating adenylate cyclase, an enzyme that regulates cAMP levels and increases protein kinase A (PKA). This, in turn, promotes NF-κB through p65 phosphorylation –κB activation, leading to a lack of CRY and reduced expression of NF-κB targets such as TNF, IL6 and CXCL1 in fibroblasts and macrophages. A similar observation has been indicated in the study where CRY1 had a strong positive correlation with the NF-κB expression level [37].

Increasing evidence indicates that the assessment of immune infiltration may predict the outcome of cancer patients. Recent studies on metastatic melanoma have shown that a high level of *BMAL1* is manifested by an increased T cell infiltration level [12]. This correlation is also shown in the current analysis. Interestingly, de Asiss et al. have shown that patients with a decreased *BMAL1* mRNA level had a worse response to anti-PD-1 therapy compared to those with the elevated *BMAL1* expression [12]. The present analysis showed a positive correlation between expression of *BMAL1* and PDCD1 in SKCM patients. The immune system is controlled by circadian rhythm and plays a significant role in the immune cells development. REV-ERB receptors regulate adhesion and motility of macrophages [38]. Rodent studies have shown that lack of *Clock* gene is manifested by a reduced number of Th1 cells. *BMAL1* and *CRY* are involved in B-cell development [39].

In silico genomic analysis, which involves the use of computational models and simulations to study genomics and gene expression, has some limitations. The existing data may be incomplete or biased, leading to potential inaccuracies in the modeling and analysis. Therefore, in silico models and simulations need to be validated and confirmed through experimental studies. However, conducting experiments to validate every aspect of the model can be time-consuming, expensive or even ethically challenging. Consequently, the validation of in silico models may be limited and the predictions or findings may need to be interpreted with caution. Biological systems are dynamic and gene expression can be influenced by various factors, including environmental stimuli and cellular context. In silico models may struggle to accurately capture the dynamic nature of biological processes, especially when the underlying regulatory mechanisms are not well understood or easily modeled.

It is important to note that the roles of circadian genes in SKCM are complex and can vary depending on various factors. The research in this area is ongoing and further studies are needed to fully elucidate their precise roles and potential therapeutic implications in melanoma.

In conclusion, the circadian rhythm study in cutaneous melanoma is an area of research that has the potential to result in new strategies for disease prevention and treatment. Disruptions in the circadian rhythm have been linked to an increased risk of cancer. It is also believed that understanding of the circadian rhythm in cutaneous melanoma could lead to new insights into the development and progression of the disease. By targeting the circadian rhythm and reducing inflammation, it might be possible to slow down the progression of cutaneous melanoma and improve patient outcomes. Moreover, the infiltration level of immune cells and their relationship with circadian rhythm in SKCM provides insights into the immune response dynamics, potential prognostic and predictive markers, optimization of treatment timing and the impact of circadian disruption on immune suppression. These findings have clinical implications for the development of targeted immunotherapies, chronoimmunotherapy strategies and the identification of biomarkers for treatment response.

## 4. Materials and Methods

### 4.1. GEPIA—Gene Expression Profiling Interactive Analysis Database

In our study, the in silico analysis was performed in relation to 24 core circadian genes (Table 1) in the samples of tissues derived from skin cancer cutaneous melanoma (SKCM) and normal skin. For the analysis, an interactive web server GEPIA was used. The GEPIA website was set up to analyze the RNA sequencing expression data of 9736 tumors and 8587 healthy tissue samples from the TCGA and the GTEx projects, using a standard processing pipeline. The GEPIA website provides customizable functions such as tumor/normal differential expression analysis, profiling according to cancer types or pathological stages, patient survival analysis, similar gene detection, correlation analysis and dimensionality reduction analysis [40,41].

In the present study, we used a tab multiple gene expression analysis. In this section, an analysis of gene expression core circadian genes SKCM using the box plot, stage and survival analysis was performed.

### 4.2. TIMER 2.0—Database

TIMER 2.0 (Tumor IMmune Estimation Resource 2.0) is a publicly available database that provides comprehensive information on the immune cell infiltration in various types of human cancers. The database integrates gene expression data from over 10.000 tumor samples from The Cancer Genome Atlas (TCGA) and the Genotype-Tissue Expression (GTEx) project, with established computational algorithms to estimate immune cell infiltration levels in the samples. This version of the webserver features immune infiltrates’ abundances estimated by multiple deconvolution methods. The platform allows comprehensive exploration of immunological, clinical and genomic features in tumors. In our analysis, we used the modules that allow to check the correlation between circadian genes’ expression and immune infiltration level of six major immune cell types: B cells, CD4+ T cells, CD8+ T cells, dendritic cells, macrophages and neutrophils. It also enables users to explore the relationship between immune cell infiltration and gene expression, as well as clinical outcomes such as survival rates and response to immunotherapy. The correlation value between gene expression and immune infiltration level was assessed by the Spearman correlation and adjusted by tumor purity. Distributions in gene expression levels are displayed using box plots, with statistical significance of differential expression evaluated using the Wilcoxon test. TIMER 2.0 is accessible free of charge at http://timer.cistrome.org/ (accessed on 21 March 2023) [26,27].

### 4.3. cBioPortal

cBioPortal (https://www.cbioportal.org/ accessed on 21 March 2023) is a web-based platform for exploring and analyzing large-scale cancer genomics data sets. It provides access to molecular data, such as DNA mutations, mRNA and microRNA expression levels, protein expression and DNA copy number alterations from over 200 cancer studies. The platform enables researchers to visualize and analyze data using a variety of tools, including interactive heatmaps, scatterplots and survival curves. The cBioPortal was developed by the Memorial Sloan Kettering Cancer Center and the Dana-Farber Cancer Institute and is freely available to researchers and clinicians around the world [42,43]. The data analysis was performed on a dataset obtained from the TCGA PanCancer Atlas, including 448 samples of Skin Cutaneous Melanoma. Figure 11 presents basic features of SCCM patients included in the analysis.

## Figures and Tables

**Figure 1 ijms-24-10140-f001:**
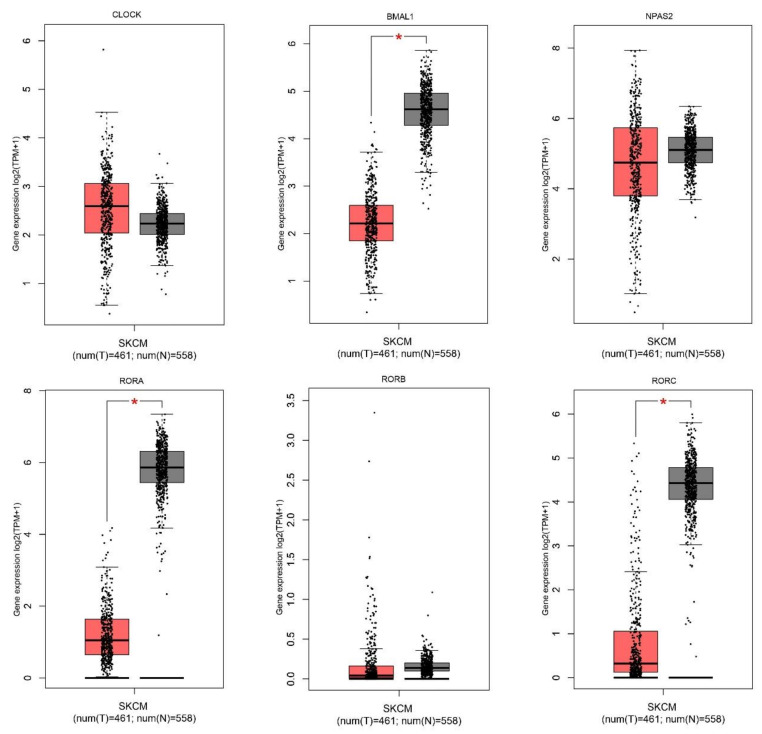
Circadian gene expression in skin cancer cutaneous melanoma (red boxes) compared to normal skin samples (grey boxes). Analysis performed by the GEPIA database using log_2_(TPM + 1) for log-scale. * *p* < 0.05.

**Figure 2 ijms-24-10140-f002:**
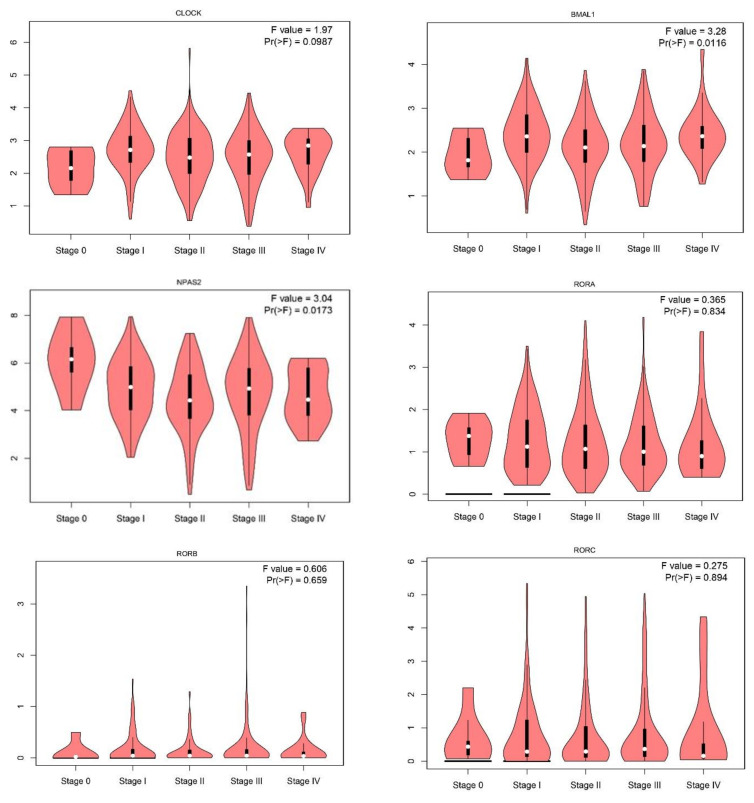
Circadian gene expression according to the stage of skin cancer cutaneous melanoma. Analysis performed by the GEPIA database using log_2_ (TPM + 1) for log-scale.

**Figure 3 ijms-24-10140-f003:**
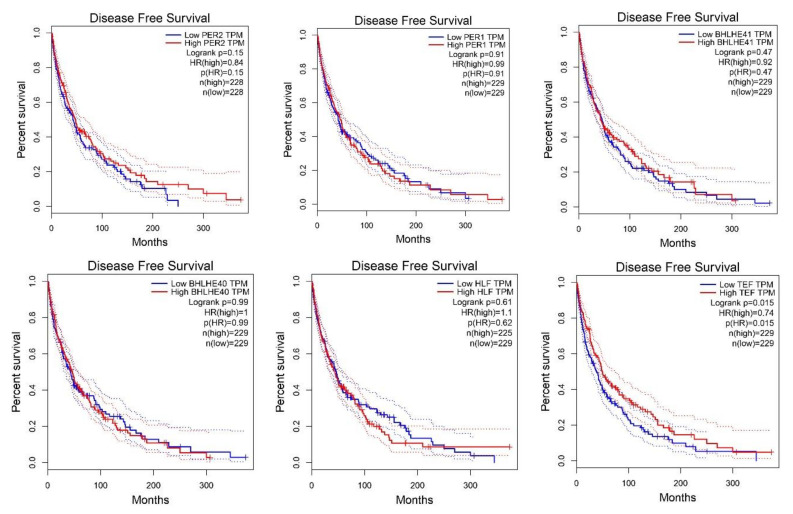
Disease-free survival in skin cutaneous melanoma patients according to low or high expression. The groups were created based on the median values. Analysis was performed by the GEPIA database. Data for *CSNK1A1L* are not available.

**Figure 4 ijms-24-10140-f004:**
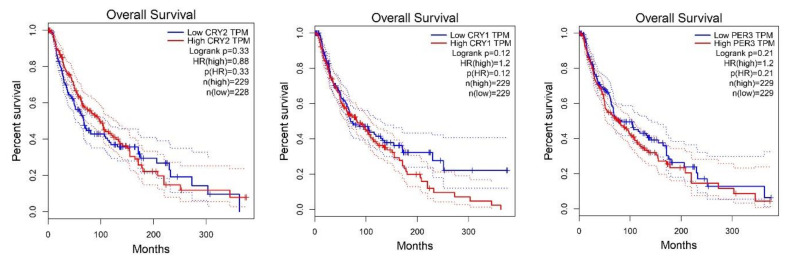
Overall survival in skin cutaneous melanoma patients according to low or high expression. The groups were created based on the median values. Analysis was performed by the GEPIA database. Data for *CSNK1A1L* is not available.

**Figure 5 ijms-24-10140-f005:**
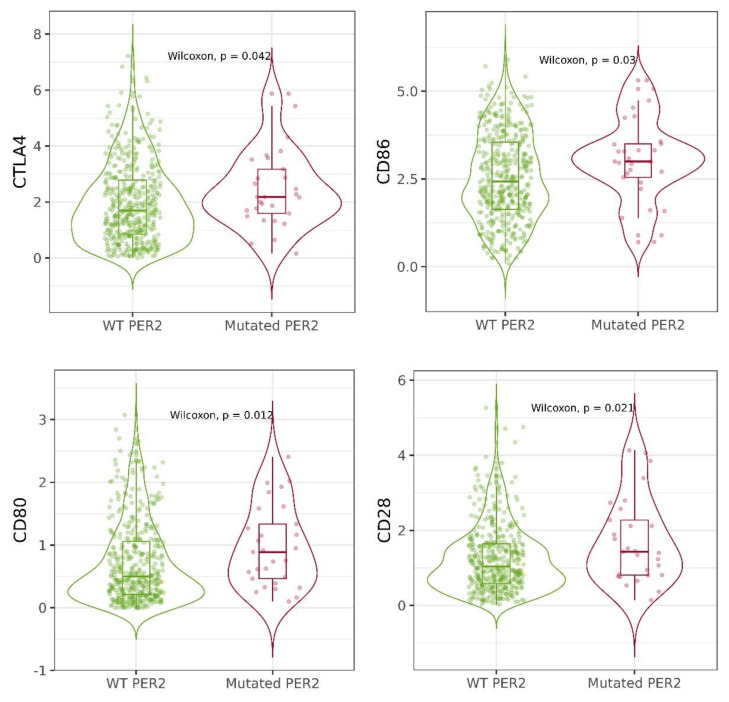
CTLA4, CD86, CD80, CD28 expression according to the mutation of PER2. Data obtained from the TIMER 2.0 database.

**Figure 6 ijms-24-10140-f006:**
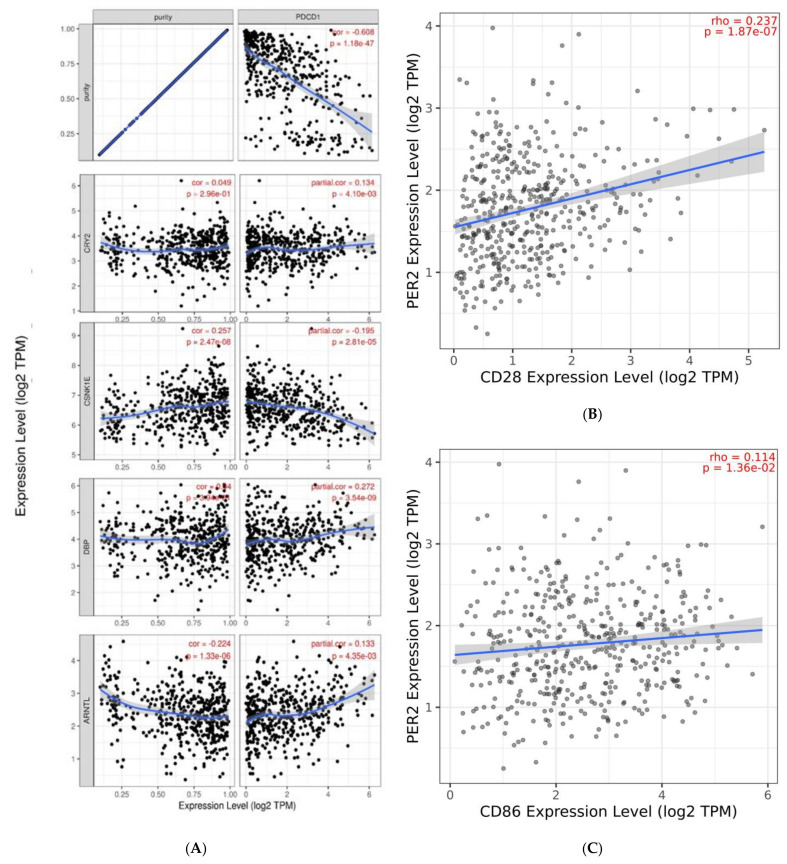
(**A**) Correlation between PDCD1 and CRY2, CSNKE1, DBP, ARNTL. (**B**) Correlation between PER2 expression and CD28 and (**C**) CD86 expression. Data obtained from the TIMER 2.0 database.

**Figure 7 ijms-24-10140-f007:**
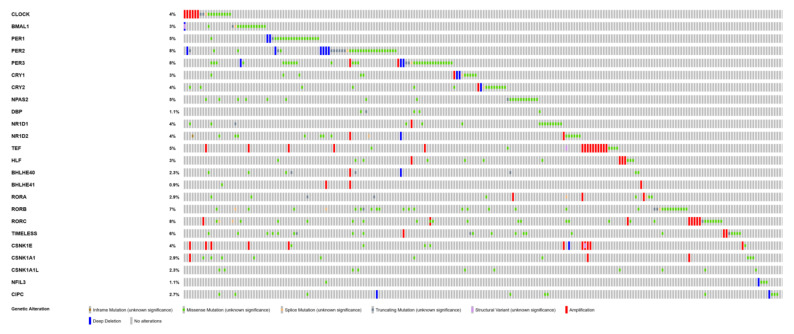
Oncoplot displaying genetic mutation landscape of core circadian genes in skin cutaneous melanoma patients. Data were obtained from the cBioPortal.

**Figure 8 ijms-24-10140-f008:**
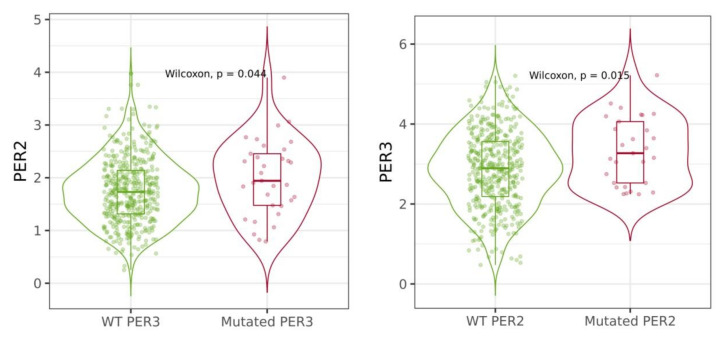
Selected circadian genes’ expression according to mutations. Data obtained from the TIMER 2.0 database.

**Figure 9 ijms-24-10140-f009:**
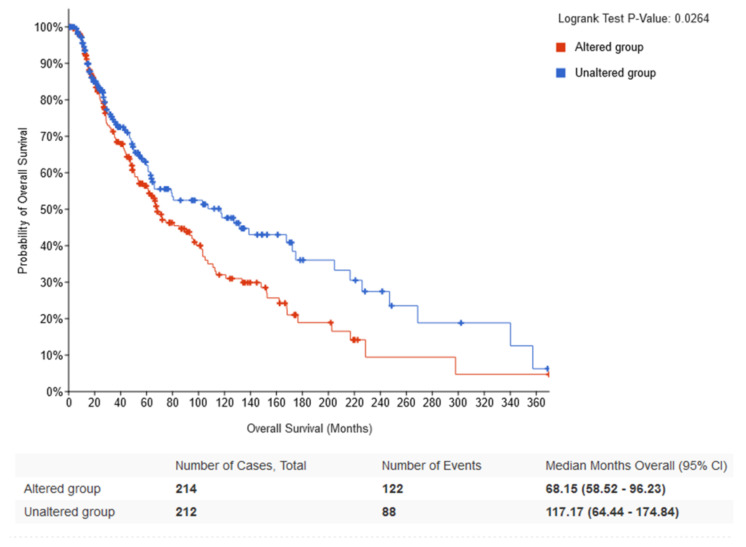
Probability of overall survival among SKCM patients, including the altered group (patients with at least one circadian gene mutation) and the unaltered group (patients without mutation of circadian genes). Kaplan Meier scatterplot obtained from the cBioPortal.

**Figure 10 ijms-24-10140-f010:**
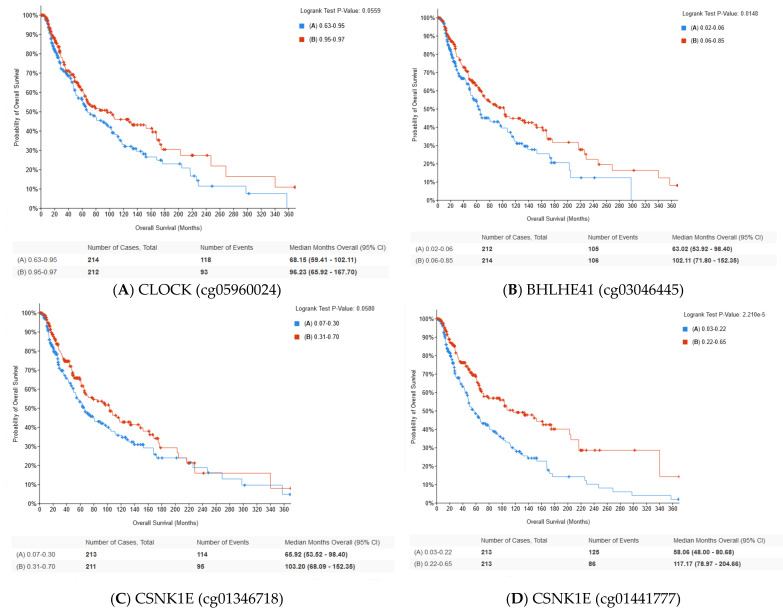
Analysis of the overall survival of selected circadian CpG site according to the cBioPortal database. Methylation level of CpG island was divided into two groups according to the median. Group A is below the median and group B above it.

**Figure 11 ijms-24-10140-f011:**
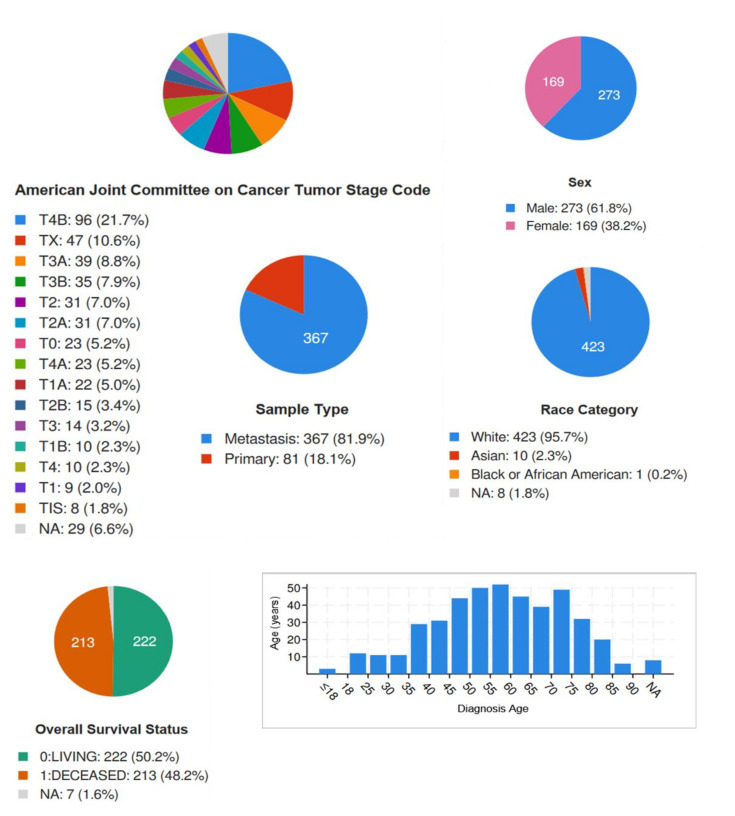
Basic patients’ characteristics obtained from the cBioPortal.

**Table 1 ijms-24-10140-t001:** Multivariate Cox regression analysis of overall survival in primary and metastatic cutaneous melanoma according to the circadian genes’ expression. The data were obtained from the TIMER 2.0 database adjusted by age, race, gender and stage of cutaneous melanoma.

	SKCM n = 471			SKCM—Primary N = 103			SKCM—Metastasis N = 368		
	HR	95% CI_l	*p*	HR	95% CI_l	*p*	HR	95% CI_l	*p*
*CLOCK*	0.90	0.757–1.081	0.27	1.13	0.685–1.849	0.64	0.89	0.733–1.086	0.26
*ARNTL*/*BMAL1*	0.71	0.562–0.889	0.00	1.16	0.643–2.108	0.62	0.68	0.529–0.877	0.00
*CRY1*	0.97	0.787–1.183	0.73	1.18	0.673–2.053	0.57	0.96	0.763–1.199	0.70
*CRY2*	0.82	0.637–1.043	0.11	1.10	0.621–1.979	0.72	0.83	0.63–1.086	0.17
*PER1*	0.93	0.775–1.107	0.40	1.03	0.591–1.796	0.92	0.93	0.769–1.128	0.47
*PER2*	0.98	0.768–1.248	0.86	1.61	0.77–3.368	0.21	0.96	0.736–1.241	0.73
*PER3*	1.00	0.849–1.176	0.99	0.80	0.456–1.391	0.42	1.02	0.856–1.217	0.82
*TEF*	0.78	0.652–0.93	0.01	0.97	0.594–1.568	0.89	0.80	0.654–0.966	0.02
*DBP*	0.86	0.720–1.03	0.1	0.84	0.486–1.46	0.54	0.88	0.719–1.077	0.21
*TIMELESS*	1.16	0.932–1.445	0.19	3.05	1.36–6.846	0.01	1.14	0.895–1.439	0.30
*RORA*	0.94	0.759–1.167	0.58	2.29	1.195–4.37	0.01	0.88	0.698–1.118	0.30
*RORB*	1.11	0.692–1.793	0.66	2.23	0.199–24.88	0.52	1.20	0.734–1.948	0.47
*RORC*	0.95	0.828–1.097	0.51	0.98	0.633–1.521	0.93	0.97	0.832–1.12	0.64
*NR1D1*	0.79	0.637–0.97	0.03	0.30	0.134–0.647	0.00	0.86	0.686–1.075	0.18
*NR1D2*	0.79	0.681–0.91	0.00	0.95	0.646–1.389	0.78	0.78	0.665–0.921	0.00
*CSNK1E*	1.17	0.922–1.493	0.19	0.62	0.311–1.244	0.18	1.26	0.97–1.636	0.08
*CSNK1A1*	0.81	0.652–0.995	0.05	1.17	0.701–1.953	0.55	0.76	0.598–0.964	0.02
*HLF*	1.05	0.826–1.336	0.69	0.93	0.42–2.076	0.87	1.04	0.804–1.346	0.77
*BHLHE40*	0.92	0.785–1.085	0.33	1.05	0.698–1.569	0.83	0.91	0.764–1.087	0.30
*BHLHE41*	0.96	0.873–1.058	0.42	0.89	0.651–1.207	0.44	0.98	0.884–1.085	0.69

**Table 2 ijms-24-10140-t002:** Correlation between circadian genes and expression levels of several cytokines and inflammatory cell genes in primary and metastatic cutaneous melanomas.

	Primary		Metastasis	
Circadian Genes		Correlation Coefficient (r) *p*		Correlation Coeffi-cient (r) *p*
*ARNTL*	*CTLA4* *IFNG* *NFKB1* *TNF*	0.20 0.043 0.21 0.032 0.61 0.000 0.20 0.044	*CTLA4* *CXCL1* *NFKB1* *IFNG* *IL6* *PDCD1* *TNF*	0.35 0.0000 0.23 0.0000 0.43 0.0000 0.34 0.0000 0.24 0.0000 0.25 0.0000 0.30 0.0000
*BHLHE40*	*CXCL1* *IL6* *NFKB1*	0.17 0.087 0.39 0.000 0.45 0.000	*CXCL1* *IL6* *NFKB1*	0.24 0.0000 0.26 0.0000 0.28 0.0000
*BHLHE41*	*NFKB1*	0.34 0.001	*CTLA4* *CXCL1* *IFNG* *NFKB1* *PDCD1* *TNF*	0.14 0.0084 0.23 0.0000 0.12 0.0271 0.15 0.0058 0.11 0.0321 0.15 0.0049
*CLOCK*	*NFKB1* *IL6*	0.61 0.000 0.22 0.022	*NFKB1*	0.61 0.0000
*CRY1*	*NFKB1* *IL6*	0.50 0.000 0.20 0.047	*NFKB1*	0.38 0.0000
*CRY2*	*NFKB1*	0.36 0.000	*NFKB1* *TNF*	0.23 0.0000 0.14 0.0066
*CSNK1A1*	*NFKB1* *IL6*	0.55 0.000 0.25 0.012	*IFNG* *NFKB1*	0.10 0.0734 0.40 0.0000
*CSNK1A1L*	*CXCL1* *IL6* *NFKB1*	0.19 0.053 0.34 0.000 0.49 0.000	*NFKB1*	0.30 0.0000
*CSNK1E*	*IL6* *NFKB1*	0.21 0.034 0.21 0.032	*NFKB1*	0.14 0.0108
*DBP*	*PDCD1*	0.18 0.062	*PDCD1* *TNF*	0.20 0.0001 0.11 0.0464
*HLF*	*IFNG* *PDCD1* *TNF*	0.19 0.058 0.18 0.069 0.20 0.048		
*NPAS2*	*CTLA4* *NFKB1*	0.34 0.001 0.27 0.006	*CTLA4* *CXCL1*	0.20 0.0001 0.22 0.0000
*NR1D2*	*NFKB1*	0.58 0.000	*IFNG* *NFKB1*	0.10 0.0610 0.45 0.0000
*PER2*	*IL6* *NFKB1* *TNF*	0.19 0.050 0.33 0.001 0.20 0.046	*NFKB1* *TNF*	0.34 0.0000 0.10 0.0592
*PER3*	*NFKB1*	0.28 0.005	*NFKB1*	0.27 0.0000
*RORA*	*NFKB1* *TNF*	0.33 0.001 0.24 0.013	*CTLA4* *IFNG* *IL6* *NFKB1* *PDCD1* *TNF*	0.26 0.0000 0.27 0.0000 0.31 0.0000 0.40 0.0000 0.20 0.0002 0.28 0.0000
*RORC*	*IFNG* *NFKB1* *PDCD1* *TNF*	0.23 0.022 0.25 0.010 0.22 0.025 0.32 0.001	*CTLA4* *CXCL1* *IFNG* *IL6* *NFKB1* *PDCD1* *TNF*	0.35 0.0000 0.13 0.0147 0.28 0.0000 0.18 0.0008 0.13 0.0175 0.29 0.0000 0.39 0.0000
*RORB*			*CXCL1* *IFNG* *PDCD1*	0.10 0.0649 0.14 0.0069 0.10 0.0634
*TEF*	*IL6* *NFKB1*	0.19 0.060 0.43 0.000	*NFKB1*	0.29 0.0000
*TIMELESS*	*CXCL1* *IL6* *NFKB1*	0.18 0.064 0.18 0.062 0.39 0.000	*NFKB1*	0.23 0.0000

**Table 3 ijms-24-10140-t003:** Correlation between circadian genes’ expression and the immune cells infiltration level in SKCM patients.

	B Cell		CD4+ T Cells		CD8+ T Cells		Dendritic Cells		Macrophages		Neutrophils	
	Correlation	*p* Value	Correlation	*p* Value	Correlation	*p* Value	Correlation	*p* Value	Correlation	*p* Value	Correlation	*p* Value
*ARNTL/* *BMAL1*	0.114	0.016	0.176	0.000	0.394	0.000	0.291	0.000	0.283	0.000	0.509	0.000
*BHLHE40*	0.095	0.045	0.053	0.266	0.140	0.003	0.137	0.004	0.121	0.010	0.240	0.000
*BHLHE41*	0.067	0.154	−0.029	0.548	0.104	0.030	0.101	0.033	0.025	0.594	0.096	0.041
*CLOCK*	0.095	0.044	0.119	0.012	0.359	0.000	0.214	0.000	0.272	0.000	0.452	0.000
*CRY1*	0.124	0.009	0.080	0.092	0.276	0.000	0.154	0.001	0.253	0.000	0.343	0.000
*CRY2*	0.138	0.003	0.372	0.000	0.170	0.000	0.207	0.000	0.257	0.000	0.257	0.000
*CSNK1A1*	−0.020	0.678	0.125	0.008	0.305	0.000	0.116	0.014	0.264	0.000	0.449	0.000
*CSNK1A1LL*	−0.026	0.587	0.059	0.213	0.170	0.000	0.074	0.120	0.148	0.002	0.280	0.000
*CSNK1E*	−0.039	0.405	0.058	0.220	−0.091	0.057	−0.046	0.330	0.052	0.271	−0.028	0.552
*DBP*	0.048	0.310	0.309	0.000	−0.022	0.649	0.136	0.004	−0.009	0.849	0.028	0.552
*HLF*	−0.045	0.343	0.161	0.001	0.064	0.182	−0.022	0.642	0.054	0.252	0.065	0.166
*NFIL3*	0.068	0.151	−0.009	0.851	0.200	0.000	0.099	0.038	0.222	0.000	0.305	0.000
*NPAS2*	0.007	0.880	−0.093	0.049	0.130	0.007	0.046	0.335	−0.036	0.449	0.088	0.062
*NR1D1*	−0.052	0.272	0.059	0.211	−0.046	0.332	−0.029	0.546	−0.094	0.045	−0.036	0.450
*NR1D2*	0.124	0.008	0.187	0.000	0.398	0.000	0.226	0.000	0.289	0.000	0.520	0.000
*PER1*	0.000	0.998	0.134	0.005	−0.007	0.880	−0.089	0.060	0.055	0.245	0.007	0.885
*PER2*	0.122	0.010	0.237	0.000	0.157	0.001	0.143	0.002	0.219	0.000	0.264	0.000
*PER3*	0.087	0.065	0.172	0.000	0.253	0.000	0.156	0.001	0.110	0.019	0.209	0.000
*RORA*	0.071	0.135	0.178	0.000	0.303	0.000	0.159	0.001	0.288	0.000	0.445	0.000
*RORB*	0.051	0.283	−0.020	0.674	0.147	0.002	0.015	0.749	−0.044	0.355	0.104	0.027
*RORC*	0.138	0.003	0.023	0.628	0.194	0.000	0.080	0.090	0.015	0.749	0.130	0.006
*TEF*	0.080	0.091	0.240	0.000	0.091	0.057	0.103	0.029	0.161	0.001	0.184	0.000
*TIMELESS*	0.148	0.002	0.059	0.215	0.149	0.002	0.165	0.000	0.091	0.052	0.198	0.000

**Table 4 ijms-24-10140-t004:** Mutations of circadian genes among SKCM samples and their frequencies among other cancer types.

Gene	Number of Mutated Samples	% of Mutated Samples	Place of Mutated Samples Among Other Cancer Types
*PER3*	37	8%	2
*PER2*	36	8%	2
*RORC*	34	8%	1
*RORB*	33	7%	1
*TIMELESS*	25	6%	2
*TEF*	22	5%	3
*PER1*	21	5%	2
*NPAS2*	20	5%	1
*CLOCK*	18	4%	4
*NR1D2*	18	4%	2
*CRY2*	17	4%	2
*CSNK1E*	17	4%	10
*NR1D1*	16	4%	1
*BMAL1*	15	3%	2
*CRY1*	14	3%	3
*HLF*	14	3%	1
*RORA*	13	3%	2
*CSNK1A1*	13	3%	3
*CIPC*	12	3%	2
*BHLHE40*	10	2%	2
*CSNK1A1L*	10	2%	5
*DBP*	5	1%	4
*NFIL3*	5	1%	8
*BHLHE41*	4	<1%	14

**Table 5 ijms-24-10140-t005:** Correlation between methylation of CpG island of circadian genes and their expression levels in SKCM patients.

CpG Island	CpG Site	Impact of Methylated CpG to mRNA Expression	Correlation Coefficient (r), *p*
*CLOCK* (cg05960024)	5′UTR	Not significant	
*ARNTL*/*BMAL1* (cg13250711)	TSS1500	Not significant	
*ARNTL*/*BMAL1* (cg06499652)	1^ST^ EXON 5′UTR	Not significant	
*BHLHE40* (cg08587820)	TSS200	Significant	r = −0.12 *p* = 0.01
*BHLHE40* (cg03223878)	1^ST^ EXON 5′UTR	Not significant	
*BHLHE41* (cg03046445)	1^ST^ EXON 5′UTR	Not significant	
*BHLHE41* (cg19243777)	TSS1500	Not significant	
*PER1* (cg16545079)	TSS200	Not significant	
*PER2* (cg05664072)	TSS1500	Not significant	
*PER2* (cg23905308)	5′UTR	Significant	r = −0.26 *p* < 0.001
*PER3* (cg06487986)	TSS1500	Not significant	
*PER3* (cg25514503)	BODY	Significant	r = −0.53 *p* < 0.001
*CRY1* (cg03742214)	1ST EXON 5′UTR	Significant	r = −0.11 *p* = 0.02
*CRY1* (cg10126874)	1ST EXON 5′UTR	Significant	r = −0.17 *p* < 0.01
*CRY2* (cg01618083)	BODY 1^ST^ EXON	Not significant	r = −0.18 *p* < 0.001
*NPAS2* (cg07799947)	TSS1500	Not significant	
*NPAS2* (cg14383135)	5′UTR	Significant	r = −0.13 *p* < 0.001
*NR1D1* (cg02337166)	TSS1500	Significant	r = −0.16 *p* = 0.009
*NR1D1* (cg22640452)	1^ST^ EXON 5′UTR	Not significant	
*NR1D2* (cg14452875	Body TSS1500	Not significant	
*TEF* (cg04137037)	Body TSS200	Not significant	
*HLF* (cg04219321)	TSS1500	Significant	r = 0.32 *p* < 0.001
*RORA* (cg13301933)	1ST EXON 5′UTR	Not significant	
*RORA* (cg27167601)	TSS1500	Significant	r = 0.23 *p* < 0.001
*RORB* (cg07536920)	1ST EXON 5′UTR	Not significant	
*RORB* (cg14331163)	TSS1500	Significant	r = −0.13 *p* = 0.06
*RORC* (cg18149207)	TSS1500	Significant	r = −0.54 *p* < 0.001
*RORC* (cg25112191)	1^ST^ EXON 5′UTR	Significant	
*TIMELESS* (cg00819696)	5′UTR	Not significant	
*TIMELESS* (cg22580905)	TSS200	Not significant	
*CSNK1E* (cg01346718)	TSS1500 5′UTR	Significant	r = −0.34 *p* < 0.001
*CSNK1E* (cg01441777)	TSS1500	Significant	r = −0.39 *p* < 0.001
*CSNK1A1* (cg05607472)	BODY	Not significant	
*CSNK1A1* (cg06899802)	TSS200	Not significant	
*CSNK1A1L* (cg01668383)	1ST EXON 5′UTR	Not significant	
*NFIL3* (cg10708828)	5′UTR	Not significant	
*NFIL3* (cg15919045)	TSS1500	Not significant	

## Data Availability

The datasets presented in this study can be found in online repositories. The names of the repository/repositories and accession number(s) can be found in the article.

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
