# Peer review of "Disruptions of Circadian Genes in Cutaneous Melanoma—An In Silico Analysis of Transcriptome Databases"

_ijms, 2023, doi:10.3390/ijms241210140_

Round 1
Reviewer 1 Report
The author analyzed the correlation of circadian genes with SKCM through the open access database, which is an interesting topic. Unfortunately, the author's research was too scattered, and did not get too many in-depth and valuable conclusions.
1. The cell composition of SKCM is completely different from that of normal skin tissue. It is worth discussing whether the conclusion drawn by using normal skin tissue as a control is convincing.
2. The abstract does not accurately summarize the main content of the article.
3. The correlation between circadian rhythm gene and tumor stage has not been further elaborated.
4. The melanoma group and the normal skin group should be clearly marked in Figure 1.
5. Insufficient resolution of Figure 7.
Author Response
Reviewer #1 The author analyzed the correlation of circadian genes with SKCM through the open access database, which is an interesting topic. Unfortunately, the author's research was too scattered, and did not get too many in-depth and valuable conclusions
- The cell composition of SKCM is completely different from that of normal skin tissue. It is worth discussing whether the conclusion drawn by using normal skin tissue as a control is convincing.
Thank you for bringing up this important point. We apologize for the lack of specific information regarding the cell types used in the RNA sequencing (RNAseq) analysis. However, it is worth noting that based on the study by Lengyel et al., the composition of skin biopsies was similar, both normal skin and tumor tissue, was reported to consist primarily of epidermal keratinocytes rather than melanocytes. Additionaly most of tumors are characterized by similar pattern of clock gene expression showing that kind of tissue did not have remarkable impact but the carcinogenic transformation is characterized by altered circadian gene expression.
- The abstract does not accurately summarize the main content of the article.
Thank you for your comment.. Abstract has been corrected.
- The correlation between circadian rhythm gene and tumor stage has not been further elaborated.
Thank you for your comment. Currently, there are limited studies comparing circadian genes according to tumor stage in cutaneous melanoma. Further research is needed in this area to better understand the relationship between circadian genes and tumor progression but we added some information in the discussion.
- The melanoma group and the normal skin group should be clearly marked in Figure 1.
Thank you for your comment.. The description of the figure 1 has been corrected.
- Insufficient resolution of Figure 7.
Thank you for your comment.. In the final version of the manuscript, we will ensure that the figures are included with improved resolution.
Reviewer 2 Report
The present study is a very interesting one acknowledging the fact that cutaneous melanoma is a major cause of skin cancer deaths and is usually diagnosed at a later stage, resulting in poor prognosis. Hence, identifying new genetic targets for therapy and prognostic biomarkers is important.
The in silico study conducted a detailed analysis of mRNA expression levels of 24 circadian genes and their correlation with immune infiltration levels in cutaneous melanoma, using multipleomics databases. The analyzed study found disruptions in circadian genes in several aspects. The in silico analysis revealed that more than half of the circadian genes investigated had significantly different expression patterns compared to normal skin.
The improvements I would suggest are concerning the degree of comprehensibility and the addressing target readers. All 24 analyzed genes should be discussed in detail and their physiological and pathological implications should be written in the article. Nevertheless, the clinical application perspective of this study should be outlines, together with the limitations of the study, even if we are facing an in silico one.
An English language review should be performed.
Author Response
Reviewer #2
The present study is a very interesting one acknowledging the fact that cutaneous melanoma is a major cause of skin cancer deaths and is usually diagnosed at a later stage, resulting in poor prognosis. Hence, identifying new genetic targets for therapy and prognostic biomarkers is important.
The in silico study conducted a detailed analysis of mRNA expression levels of 24 circadian genes and their correlation with immune infiltration levels in cutaneous melanoma, using multipleomics databases. The analyzed study found disruptions in circadian genes in several aspects. The in silico analysis revealed that more than half of the circadian genes investigated had significantly different expression patterns compared to normal skin.
The improvements I would suggest are concerning the degree of comprehensibility and the addressing target readers. All 24 analyzed genes should be discussed in detail and their physiological and pathological implications should be written in the article. Nevertheless, the clinical application perspective of this study should be outlines, together with the limitations of the study, even if we are facing an in silico one
Thank you for providing your feedback on the article. We appreciate your comments and suggestions. However, we would like to address some concerns regarding the feasibility and scope of implementing all the suggested improvements.
While we understand the importance of discussing all 24 analyzed genes in detail, it's essential to consider the limitations of space and readability within the article. A comprehensive discussion of each gene may lead to an excessively lengthy manuscript, which could hinder the overall flow and readability for the target readership. To maintain clarity and conciseness, we have focused on highlighting the most significant findings and their implications for the study.
Additionally, we have included a section explicitly addressing the limitations of our in silico study, highlighting potential areas for further research and validation.
Thank you again for your valuable feedback, which contributes to the overall quality of the manuscript.
Round 2
Reviewer 1 Report
The author has compiled an encyclopedia like summary of CRP related content, from molecular medicine to pathophysiology further to clinical diagnostics. However, this review is too long and has poor readability for readers. Moreover, as a widely used but not highly targeted laboratory indicator in clinical practice, CRP is usually used as a basic test rather than a specific test. Therefore, it is recommended to simplify the content related to CRP and clinical diseases, as it is difficult to summarize a clear conclusion and has limited clinical significance.
Author Response
I recently came across your response to my article, and I would like to address a concern I have regarding a possible mistake that might have occurred.
While I appreciate your feedback and engagement with the topic, it seems that you may have inadvertently referred to a different article or study in your response. The points you raised do not align with the content of my article, and it has left me puzzled as to whether there was a miscommunication or an oversight.
I kindly request you to review my article once again and provide specific points or arguments that you believe require further clarification or improvement. I value your expertise and believe that constructive criticism can significantly contribute to the quality of the work.
I look forward to your revised assessment and the opportunity to address any valid concerns you may have. Thank you for your time and consideration
Reviewer 2 Report
Despite the fact that the changes performed and the information added are not at the highest solicited level, I agree with the publication of the manuscript in this form.
Author Response
Thank you for accepting our article and for your valuable comment. We have taken your suggestion into consideration and have decided to move Table 1 to the supplemental materials. This adjustment aims to enhance the readability of the article and make it easier for readers to engage with the core content. We believe that this modification will contribute to a more seamless reading experience for our audience